# The “Hygiene Package”: Analysis of Fraud Rates in Italy in the Period before and after Its Entry into Force

**DOI:** 10.3390/foods11091244

**Published:** 2022-04-26

**Authors:** Annalisa Previti, Domenico Vicari, Francesca Conte, Michela Pugliese, Valeria Gargano, Angela Alibrandi, Agata Zirilli, Annamaria Passantino

**Affiliations:** 1Department of Veterinary Sciences, University of Messina, 98122 Messina, Italy; annalisa.previti@yahoo.it (A.P.); fconte@unime.it (F.C.); passanna@unime.it (A.P.); 2Istituto Zooprofilattico della Sicilia “A.Mirri”, 90129 Palermo, Italy; domenico.vicari@izssicilia.it (D.V.); valeria.gargano@izssicilia.it (V.G.); 3Unit of Statistical and Mathematical Sciences, Department of Economics, University of Messina, 98122 Messina, Italy; angela.alibrandi@unime.it (A.A.); agata.zirilli@unime.it (A.Z.)

**Keywords:** food fraud, adulteration, inspection activities, Italy, legislation

## Abstract

In violation of EU legislation, fraudulent activities in agri-food chains seek to make economic profits at the expense of consumers. Food frauds (FFs) often constitute a public health risk as well as a risk to animal and plant health, animal welfare and the environment. To analyze FFs in Italy during 1997–2020 with the aim of gaining observational insights into the effectiveness of the legislation in force and consequently of inspection activities, FFs were determined from official food inspections carried out by the Central Inspectorate of Quality Protection and Fraud Repression of Agri-food Products in 1997–2020. Inspected sectors were wine, oils and fats, milk and dairy products, fruit and vegetables, meat, eggs, honey, feeds and supplements, and seeds. Data show that the inspection activities have significantly improved in terms of sampling and fraud detection. However, a higher incidence of fraud involving the meat sector was observed. The obtained results demonstrate that there has not been a clear change of direction after the so-called “hygiene package” (food hygiene rules in the EU) came into force. Thus, more effective measures are needed to manage risk as well as new analytical solutions to increase the deterrence against meat adulteration and the rapid detection of fraud.

## 1. Introduction

Fraudulent activities that violate EU agri-food chain legislation, characterized by their intentional nature, may constitute a risk to human, animal or plant health, to animal welfare or the environment. They include deliberate and intentional substitution, addition, adulteration or misrepresentation of food, food ingredients or packaging, or misleading claims about a product, for an unauthorized economic gain [1], increasing the apparent value of a product by intentionally misleading consumers or lowering production costs by using cheaper substitutes or even non-food-grade ingredients to undercut competitors [2].

Robson et al. [3] have highlighted that definitions and types of food fraud (FF) often differ, thereby causing confusion and increasing ambiguity due to the implementation of generalized and non-specific prevention/mitigation strategies to a particular supply chain. However, irrespective of the definition used, it is the same problem that is being addressed, namely, fraud against consumers for economic gain. Various FFs have occurred in several countries around the world [4,5,6]. The EU considers several types of FFs that can appear alone or in combination [7], as represented in Figure 1.

The European Union (EU) has not legally defined FF, resulting in the creation of varying definitions from regulatory bodies, although EU food law lays down some relevant provisions concerning FF (as shown in Table 1), recognizing the importance of FF as a risk factor.

In the EU, four criteria are used to determine whether a case should be considered as a case of food fraud or non-compliance: (i) the violation of EU rules relating to food safety; (ii) the deception of the consumer/customer (i.e., changing the labelling of a product); (iii) financial gain that is a direct or indirect economic advantage for the perpetrator; and, finally (iv) the intention, for example, to replace high-quality ingredients (or parts of ingredients) with others of lower value [11].

If food is intentionally modified by individuals or groups to harm consumers or for purposes of making economic profits this constitutes a food crime [12,13,14]. Both cases (harm and economic profit/gain) may raise questions about food safety and/or food quality and play a major role in negatively impacting consumers’ trust in food industries and government agencies [13]. A food crime may occur at any phase of food production, from processing to retailing or distribution.

To tackle and deter fraudulent practices in the agri-food chain, the EU has created the EU Food Fraud Network, which authorizes member states to exchange information and to cooperate on a voluntarily basis with respect to FF [15]. Furthermore, member states have individually instituted regulatory bodies to defend themselves against FF. An example of this is the Central Inspectorate of Quality Protection and Fraud Repression of Agri-food Products (hereinafter ICQRF) established in Italy in 1986 as part of the Italian Agricultural, Food and Forestry Policies, operating throughout the national territory to counter fraudulent practices and monitor regulatory interventions.

The ICQRF has an inspection system covering the whole country, with inspection activities performed in all agri-food sectors, from production through processing and distribution to retailing.

This study analyzes the phenomenon of FF in Italy over the period 1997–2020 through monitoring activities in different agri-food sectors carried out by ICQRF with the scope of confirming whether EU regulations—the so-called “hygiene package” including Regulations EC 852/2004 on the hygiene of foodstuffs, 853/2004 setting out specific hygiene rules for food of animal origin, 854/2004 laying down rules for the organization of official controls and 882/2004 on official controls to ensure the verification of compliance—have ensured an adequate level of consumer protection, especially in 2020, during the COVID-19 pandemic—the theatre of an exceptional crisis in which FF may have severe socio-economic implications.

## 2. Materials and Methods

To analyze the phenomenon of agri-food fraud in this study, the definition of FF suggested by Spink and Moyer [16] was adopted, as said above (Section 1, Introduction). According to this definition, an FF occurs whenever an inspected product features any kind of irregularity, no matter if it is of an administrative or a criminal nature or whether it leads to the confiscation of the product or other administrative penalties such as fines and warnings.

### 2.1. Data Collection

The utilized data are related to the number of inspection activities against fraud carried out by ICQRF of the Italian Ministry of Agriculture and to irregularities in all the different agri-food sectors over the period 1997–2020. All data were taken from annual reports, available from the website https://www.politicheagricole.it/flex/cm/pages/ServeBLOB.php/L/IT/IDPagina/394 (accessed on 11 October 2021), and were inserted into a database in Excel that systematically recorded all ICQRF inspection activities disaggregated per sector level and year. Sectors of the inspected products were the following: wine, oils and fats, milk and dairy products, fruit and vegetable, meat, eggs, honey, feeds and supplements, and seeds.

Since there was not always clear and exhaustive documentation, the following survey is to be considered reliable insofar as it reveals general trends. The analysis was conducted—within the limits allowed by the amount of data available to us—in the most neutral and objective way possible.

### 2.2. Statistical Analysis

“Dispute rates” were calculated comparing the number of irregularities to the total number of inspections carried out in each sector for each year of the series examined.

Boxplots were realized to show dispute rates in each sector comparing two periods (pre- and post-2004) in relation to the coming into force of the “hygiene package” in 2004.

To assess the existence of possible significant differences in dispute rates between the first period (1997–2004, pre-2004) and the second period (2005–2020, post-2004) for each food sector, the Non-Parametric Combination (NPC) procedure was applied [16].

The NPC Ranking procedure [17,18] was applied to the data divided into two periods, pre- and post-2004. It is a non-parametric statistical method of aggregation that provides a combined final ranking. For each year, a partial ranking of the food sectors was drawn up (based on the fraud rate); subsequently, to summarize the partial annual rankings in a global ranking (both for the period before 2004 and after 2004), the Fisher combination function was applied. Horizontal bar graphs were realized to visualize the global rankings of the product sectors before and after 2004.

Finally, the trend index was calculated to quantify the variations (increases or decreases) in fraud rates in the whole considered period. It is a measure of the evolution over time of the phenomenon under examination and includes, for every sector, all the dispute rates manifested in all examined years.

A *p*-value lower than 0.050 (two-sided) was considered statistically significant and highlighted in bold.

Statistical analyses were performed using the SPSS for Windows package, version 22.0.

## 3. Results

A total of 78,778 irregularities was detected for 444,007 inspections (in different agri-food-sectors) (Table 2).

Table 3 shows the dispute rates in each sector for each year of the examined series.

Figure 2 shows the boxplots of dispute rates for each foods sector in the two examined periods (before and after 2004).

The descriptive statistics (means ± standard deviation and median) for dispute rates and for each food sector were calculated for both the first period (1997–2004) and the second period (2005–2020). Comparison between two periods, for each food sector, was performed by means of the NPC procedure and the *p*-value is reported (Table 4). The hypotheses system is formalized as follows:(1)$$H_{0}:\left\{{Dispute\;rates\;wine}_{1}\;\overset{d}{=}{Dispute\;rates\;wine}_{2}\right\}\cap \ldots \cap \left\{{Dispute\;rates\;meat}_{1}\;\overset{d}{=}{Dispute\;rates\;meat}_{2}\right\}$$

Against
(2)$$H_{1}:\left\{{Dispute\;rates\;wine}_{1}\;\overset{d}{\neq }{Dispute\;rates\;wine}_{2}\right\}\cup \ldots \cup \left\{{Dispute\;rates\;meat}_{1}\;\overset{d}{\neq }{Dispute\;rates\;meat}_{2}\right\}$$

*d* identifies the “inequality in distribution”, 1 identifies the first period under examination (pre 2004) and 2 identifies the second period under examination (post 2004).

Examining Table 4, we can note that statistically significant differences were observed. Relating to “wine”, “honey” and “sowing seeds” sectors, dispute rates before 2004 were significantly higher than in the subsequent period (*p* = 0.001, *p* = 0.047, and *p* = 0.001, respectively); on the contrary, the “meat” sector before 2004 presented dispute rates significantly lower than the subsequent period (*p* = 0.001); the other sectors, “milk and dairy products”, “oils and fats”, “feed and supplements”, “eggs” and “vegetable preserves”, show slight variations in the rates for the two periods which are not statistically significant (*p* > 0.050).

The NPC Ranking procedure allowed us to realize, for each year, partial rankings of the food sectors (based on fraud rates) and to summarize them in two global rankings (before and after 2004, respectively) by the Fisher combination function. Table 5 reports the ordering obtained by NPC ranking application related to the fraud rates for the various sectors; the two global rankings show a notable difference in position, in the pre- and post-2004 rankings, of the fraud rate found in the meat sector, which went from ninth to second place. In addition, the honey sector occupies widely different positions in the two periods, passing from fourth to seventh place, thus indicating an increase in the fraud rate, contrary to what was found in the meat sector. In other sectors, the changes in position in the two rankings are not significant.

From a strictly statistical point of view, it should be noted that the use of the Fisher combination function, envisaged by the NPC ranking methodology, does not provide dissimilar results to those that emerged from the average values of the fraud rates in the years of the two periods examined.

Figure 3 and Figure 4 illustrate, by means of horizontal bar graphs, the global rankings of the product sectors for the two investigated periods; they show the values obtained from the application of the Fisher combination function (a fundamental tool in the NPC ranking), which summarizes the fraud rates in the years before and after 2004, separately. The values obtained with the Fisher combination function are expressed using the same unit of measure as the fraud rates (part-to-whole ratios).

Finally, the application of the trend index highlights variations in fraud rates recorded in the examined period (1997–2020) (Table 6).

The obtained results show that, for the whole period (1997–2020), the wine, feed and supplements, and honey sectors saw an almost imperceptible decrease in fraud rates, unlike what can be observed for sowing seeds, for which the reduction in value (−4.266) assumes greater importance. This negative value indicates that, in the entire examined period (1997–2020), the fraud rate in this sector showed a significant decrease. Despite the reduction in the fraud rates in these food sectors, it is to be highlighted that the value found for meat continues to show an increase, representing the maximum peak of the whole series. For the other items, a similarity is observed in the increased values ranging from 2.265 to 3.152.

## 4. Discussion

This study has revealed a higher incidence of irregularity involving the meat sector (sale of meat preserves obtained from animals of species other than those declared, violations relating to the system of labeling and presentation of meat due to omission of mandatory indications, non-compliant use of the sales denomination, irregularities in the indication of the expiry date, use of misleading terms). This result is confirmed by another study [18] in which beef counterfeits were found to represent 42.9% of all cases identified. This percentage is based on notifications relating to food products published in the Rapid Alert System for Food and Feed (RASFF) and HorizonScan in the period 1997–2017. This includes products from unapproved premises and products without inspection or found with fraudulent or missing documentation, such as certificates of entry or health certificates (falsified or fraudulent documents). This suggests that illegal manufacturing accounted for the majority of frauds identified in the two decades that were examined. It would also seem that the increase in disputes after 2004 is in accordance with the increase in reports which took place in 2013 [19], the year of the scandal concerning horse meat in some food products and not declared on the label. The European Union responded by providing for the implementation of a second European control plan targeting the prevalence of fraudulent practices in the marketing of certain food products with the European Commission Recommendation 2014/180/EU [20].

The most obvious consequence of these episodes has been a loss of consumer confidence, especially in recent years due to the recent scandals in which horse meat was added to meat products instead of beef [21]. In line with the increase in irregularity in the meat sector, these products are responsible for the worst impacts on confidence among consumers [22]. Furthermore, it has been shown that lack of consumer confidence is directly proportional to the degree of transformation of the food. For this reason, olive oil has been considered one of the food products that sustains more trust. Nonetheless, olive oil, along with honey, is one of the most falsified foods [22], as reported in this study. For example, irregularities found in the honey sector were: production, holding for sale or marketing of single-flower honeys of botanical origin and organoleptic characteristics not meeting the category declared; marketing of honeys with composition characteristics not complying with the legal parameters; and infringements relating to the labelling and/or packaging system by omission of mandatory indications, use of misleading phrases, or incorrect indications of the minimum storage period or sales name.

Additionally, specific areas in the supply chain deserve attention to achieve a greater level of consumer protection. Indeed, Robson et al. [19] report that 36.4% of the frauds are related to primary processing, of which 95.5% are cases of counterfeiting involving products manufactured/packaged in unapproved premises or without adequate inspection or documentation, as well as products issued with fraudulent health certificates. Considering the increase in these categories of fraud despite the introduction of the Regulation (EC) No. 852/2004, it could be deduced that the means of prevention associated with this legislation are ineffective and not implemented.

The first step in preventing food fraud is to determine an ingredient’s potential vulnerability to fraud by examining factors known to help predict the occurrence of fraud [23]. The way forward to achieve a reduction in food fraud is to develop a strong control culture and new risk-management strategies [24,25]. The main programs implemented to date to ensure food safety are the Six Sigma and Hazard Analysis and Critical Control Point methodologies. These programs are closely related to good agricultural practices, good production practices and hygiene standard operating procedures along production chains. Another strategy to reduce food fraud is to develop industry standards and certifications (for example, the Global Food Safety Initiative) and to increase the identification of incidents. This could be achieved by collecting results in databases such as USP [26].

Innovations in testing methods (e.g., USP and the Food Integrity Project of the European Commission), expansion of law enforcement activities (e.g., Europol/INTERPOL Operation OPSON) and introduction of new laws (e.g., US Food Safety Modernization Act, Chinese Food Safety Laws, etc.) will also contribute to reducing food fraud and adulteration [4].

Certainly, other factors independent of the limits of legislation may be implicated in the increase in fraud affecting the meat sector.

Among these is the presumable increase in the consumption of meat, which until the last century was almost non-existent on the tables of Italians, whose diet was mainly based on poor foods such as legumes [27].

Market globalization has certainly favoured criminal activities linked to fraud, as it has favoured criminal activities that revolve around longer and more complex markets often characterized by anonymous suppliers, extending the impact of these criminal activities to a greater number of consumers [16]. In Italy, food frauds are often attributable to criminal activities that may be associated with organized criminal structures, such as the mafia, Cosa Nostra, Ndrangheta, etc. When organized, food frauds are punished in accordance with the article 474*ter* of the Italian penal code on the aggravating circumstances in the trade of counterfeit products, in which counterfeiting is charged with more severe penalties if organized and committed in a systematic way.

Finally, the evolution of biotechnological knowledge has favoured the development of new methodologies aimed at altering, reducing, or slowing down unwanted conditions and processes in foodstuffs or to give them characteristics that do not naturally occur.

Irregularity in the food supply system could be exacerbated by shortages caused by climate change and the impact of COVID-19, as highlighted by other authors [28].

The reasons for the increased incidence of fraud during the pandemic should be examined, as outlined in the guidelines of the World Health Organization and the United Nations Food and Agriculture Organization in the rush to identify new suppliers due to the lack of availability or to receive supplies. This may lead companies to pay less attention to the integrity of the food chain, especially given the temporary suspension or reduction of controls by the authorities. Security risks have prompted the application of measures to reduce food supply, negatively impacting food availability [29]. Additionally, COVID-19 has raised significant food and nutrition security issues around the world which most likely have led to increased poverty, food fraud and restricted food supply and access [28]. Accelerated investments to develop more inclusive, sustainable and resilient food systems will help reduce the impact of the pandemic and provide a way to control the anticipated food security crisis and economic growth [29].

For all the other sectors, given the reduction in the number of complaints observed, it is presumable that the legislation was effective.

This study has shown that the main target of food fraud and adulterations is the meat sector, which undoubtedly appears to be the most vulnerable and therefore requires adequate strategies to identify and prevent fraud to guarantee greater consumer protection. Cases of food fraud highlight significant weaknesses in supply chain transparency and the traceability of raw meat [30]. Therefore, more effective measures are needed to manage risk along with new analytical solutions to increase the deterrence against meat adulteration and the rapid detection of fraud. It would seem, in fact, from the statistical analysis carried out, that there has not been a clear change of direction after the hygiene package came into force (in the year 2004). Research has certainly already moved towards the implementation of DNA-based methods, such as the application of the new LCD array and next-generation sequencing (NGS) to detect the adulteration of large meat species. Still, new technologies such as rapid evaporative ionization mass spectrometry (REIMS) are showing promising analytical approaches for the rapid detection of various malpractices [30].

Tackling the problem of food fraud, therefore, also means improving stringent regulatory systems, guaranteeing greater sampling and monitoring systems, training food producers and managers and developing effective, rapid, and economical methods of fraud detection.

Complex food supply chains are most at risk of fraud and special measures should be put in place within these sectors. Monitoring results can be used to analyze food safety risks to protect consumer health and rights and to prioritize target areas for food research and policymaking to enforce food safety standards [4].

Moreover, the obtained results show that, for the whole period (1997–2020), the honey sector presented an almost imperceptible phase of decrease in fraud rates, unlike what appears in other studies conducted in the EU [31,32]. Richter et al. [33] reported the addition of honeydew and walnut honey to bee honey as one of the most frequent frauds involving honey. It seems that lack of confidence is directly proportional to the degree of food processing. Thus, olive oil has been considered one of the foodstuffs that sustains more confidence among respondents. Although olive oil—together with honey—is one of the most falsified foodstuffs today [31,32], our analysis showed that wine and olive oil production are the most inspected agri-food activities, given their larger economic sizes and their higher exposure to fraud.

## 5. Conclusions

Ultimately, COVID-19 has raised fleeting food and nutrition security problems around the world which most likely have led to increased poverty, food fraud and restricted food supply and access. Accelerated investments to develop more inclusive, sustainable, and resilient food systems will help reduce the impact of the pandemic and therefore provide a way to control the anticipated food security crisis and economic growth.

## Figures and Tables

**Figure 1 foods-11-01244-f001:**
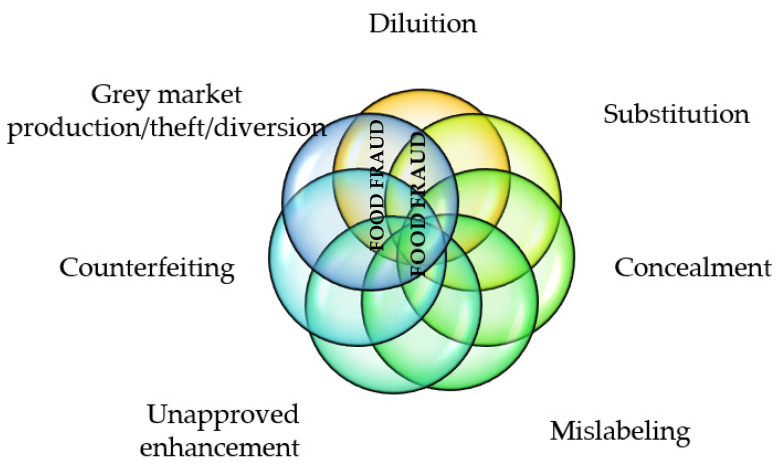
Types of food fraud. Adapted from https://knowledge4policy.ec.europa.eu/food-fraud-quality/topic/food-fraud_en (access on 14 November 2021).

**Figure 2 foods-11-01244-f002:**
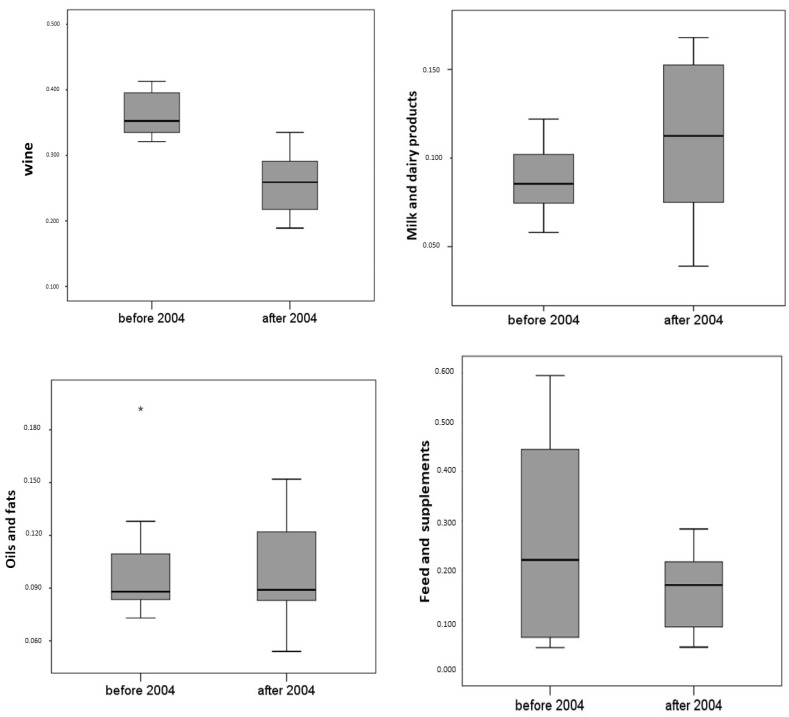
Boxplots of dispute rates for each foods sector.

**Figure 3 foods-11-01244-f003:**
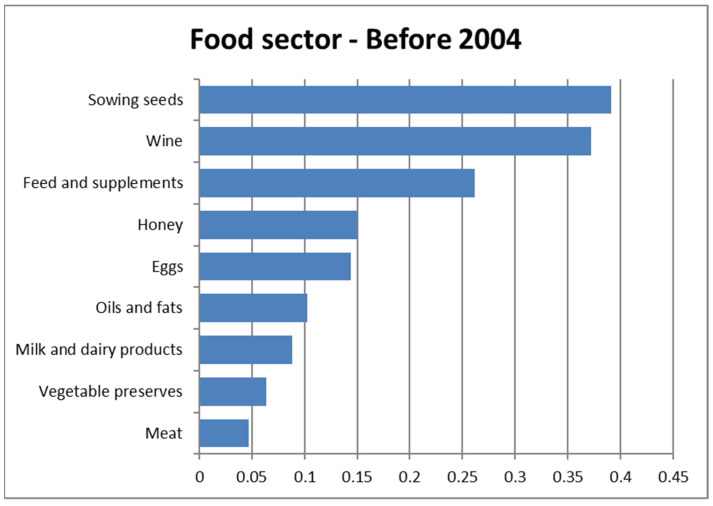
Ranking of food sectors before 2004.

**Figure 4 foods-11-01244-f004:**
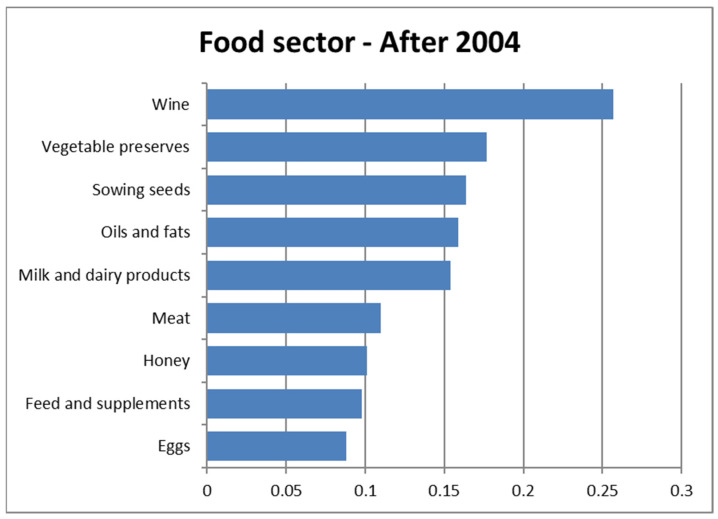
Ranking of food sectors after 2004.

**Table 1 foods-11-01244-t001:** EU legal previsions on food fraud.

EU Law	Articles	Previsions
Regulation (EC) 178/2002 [8]	Art. 8 (Protection of consumers’ interests)	Food law shall aim at the protection of the interests of consumers and shall provide a basis for consumers to make informed choices as to the foods they consume. It shall aim at the prevention of: (a) fraudulent or deceptive practices; (b) the adulteration of food; and (c) any other practices which may mislead the consumer.
Regulation (EU) 1169/2011 [9]	Art. 7, point 1 (Fair information practices)	Food information shall not be misleading, particularly: (a) as to the characteristics of the food and, in particular, as to its nature, identity, properties, composition, quantity, durability, country of origin or place of provenance, method of manufacture or production (b) by attributing to the food effects or properties which it does not possess; (c) by suggesting that the food possesses special characteristics when in fact all similar foods possess such characteristics, in particular, by specifically emphasizing the presence or absence of certain ingredients and/or nutrients; (d) by suggesting, using the appearance, the description or pictorial representations, the presence of a particular food or an ingredient, while in reality a component naturally present or an ingredient normally used in that food has been substituted with a different component or a different ingredient.
Regulation (EU) 2017/625 [10]	Art. 9, points 1 and 2 (General rules on official controls)	1. Competent authorities shall perform official controls on all operators regularly, on a risk basis and with appropriate frequency, taking account of: […] any information indicating the likelihood that consumers might be misled, in particular as to the nature, identity, properties, composition, quantity, durability, country of origin or place of provenance, method of manufacture or production of food. […] 2. Competent authorities shall perform official controls regularly, with appropriate frequencies determined on a risk basis, to identify possible intentional violations of the rules […] perpetrated through fraudulent or deceptive practices, and taking into account information regarding such violations shared through the mechanisms of administrative assistance.

**Table 2 foods-11-01244-t002:** Number of irregularities detected in different agri-food-sectors.

Agri-Food Sectors	Number of Inspections	Number of Irregularities
**Wine**	159,187	44,836
**Milk and dairy products**	67,413	6701
**Oils and fats**	81,790	7883
**Feed and supplements**	36,361	5576
**Eggs**	17,784	2582
**Honey**	12,224	1209
**Sowing seeds**	16,061	3738
**Vegetable preserves**	23,139	1764
**Meat**	30,049	4489
	**444,007**	**78,778**

**Table 3 foods-11-01244-t003:** Dispute rates for sector and year.

Year	Wine	Milk and Dairy Products	Oils and Fats	Feed and Supplements	Eggs	Honey	Sowing Seeds	Vegetable Preserves	Meat
**1997**	0.321	0.081	0.086	0.391	0.138	0.233	0.585	0.071	0.037
**1998**	0.346	0.058	0.073	0.499	0.166	0.188	0.461	0.055	0.024
**1999**	0.359	0.084	0.091	0.594	0.197	0.126	0.481	0.031	0.015
**2000**	0.341	0.068	0.128	0.322	0.084	0.233	0.502	0.061	0.010
**2001**	0.413	0.087	0.083	0.121	0.137	0.171	0.243	0.142	0.002
**2002**	0.496	0.114	0.084	0.061	0.131	0.123	0.271	0.050	0.028
**2003**	0.329	0.122	0.090	0.069	0.115	0.038	0.238	0.050	0.152
**2004**	0.378	0.090	0.192	0.044	0.189	0.093	0.347	0.059	0.112
**2005**	0.267	0.065	0.085	0.054	0.143	0.094	0.134	0.044	0.055
**2006**	0.191	0.086	0.072	0.045	0.116	0.065	0.109	0.039	0.052
**2007**	0.239	0.084	0.089	0.054	0.079	0.058	0.098	0.071	0.059
**2008**	0.330	0.123	0.084	0.066	0.106	0.121	0.139	0.135	0.113
**2009**	0.274	0.160	0.130	0.121	0.095	0.117	0.431	0.093	0.196
**2010**	0.277	0.165	0.094	0.148	0.207	0.078	0.275	0.136	0.268
**2011**	0.335	0.148	0.128	0.198	0.323	0.149	0.255	0.114	0.296
**2012**	0.324	0.157	0.123	0.235	0.240	0.099	0.068	0.152	0.361
**2013**	0.305	0.137	0.121	0.284	0.163	0.124	0.066	0.135	0.300
**2014**	0.204	0.120	0.082	0.231	0.169	0.118	0.186	0.084	0.181
**2015**	0.218	0.168	0.084	0.262	0.207	0.132	0.167	0.089	0.174
**2016**	0.217	0.105	0.112	0.191	0.17	0.103	0.168	0.104	0.234
**2017**	0.254	0.072	0.089	0.162	0.112	0.087	0.123	0.076	0.105
**2018**	0.221	0.039	0.054	0.204	0.119	0.106	0.122	0.056	0.194
**2019**	0.264	0.078	0.152	0.179	0.101	0.1	0.207	0.041	0.134
**2020**	0.189	0.051	0.077	0.106	0.115	0.046	0.068	0.038	0.116

**Table 4 foods-11-01244-t004:** Descriptive statistics of dispute rates and *p*-values related to comparisons between two periods.

Food Sector	Before 2004		After 2004		*p*-Value
	Mean ± S.D.	Median	Mean ± S.D.	Median	
Wine	0.372 ± 0.057	0.352	0.257 ± 0.049	0.259	**0.001**
Milk and dairy products	0.088 ± 0.021	0.085	0.110 ± 0.043	0.112	0.284
Oils and fats	0.103 ± 0.039	0.088	0.098 ± 0.026	0.089	0.927
Feed and supplements	0.262 ± 0.217	0.221	0.159 ± 0.078	0.170	0.444
Eggs	0.144 ± 0.037	0.137	0.154 ± 0.065	0.131	0.976
Honey	0.150 ± 0.068	0.148	0.101 ± 0.028	0.102	**0.047**
Sowing seeds	0.391 ± 0.133	0.404	0.164 ± 0.095	0.137	**0.001**
Vegetable preserves	0.064 ± 0.033	0.057	0.088 ± 0.039	0.087	0.188
Meat	0.047 ± 0.054	0.026	0.177 ± 0.095	0.178	**0.001**

**Table 5 foods-11-01244-t005:** NPC ranking for food sectors before and after 2004.

Food Sectors	Before 2004	After 2004
Wine	2	1
Milk and dairy products	7	6
Oils and fats	6	8
Feed and supplements	3	4
Eggs	5	5
Honey	4	7
Sowing seeds	1	3
Vegetable preserves	8	9
Meat	9	2

**Table 6 foods-11-01244-t006:** Trend indices of the fraud rates in the food sectors for the period 1997–2020.

Food Sectors	1997–2020
Wine	−0.304
Milk and dairy products	2.734
Oils and fats	2.265
Feed and supplements	−0.936
Eggs	2.394
Honey	−0.994
Sowing seeds	−4.266
Vegetable preserves	3.152
Meat	8.757

## Data Availability

Data are available contacting the corresponding author (michela.pugliese@unime.it).

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
