# Peer review of "The “Hygiene Package”: Analysis of Fraud Rates in Italy in the Period before and after Its Entry into Force"

_foods, 2022, doi:10.3390/foods11091244_

Round 1

Reviewer 1 Report

The authors reported a study that shows the food fraud cases in Italy from 1997 to 2020. There are some serious issues regarding this manuscript for the authors to address:

  • The authors should be more careful throughout the manuscript. For example, there is a problem labelling the figures. Also, the abstract includes the words “background”, objective, etc.
  • The paper needs more references and examples in order to explain the food fraud in each sector.
  • The hygiene package should be explained in more detail.

Author Response

The manuscript was strongly revised in accordance with the suggestions of reviewers. 

Reviewer 2 Report

- Although there is statistical support for analytical data on reports of food fraud, data on the total number by years and categories are not expressed. Recommendation - this information would be interesting to readers if they are presented in tabular form, with total numbers of fraud notifications by year and category.
- The Conclusions mention only one single modern analytical technique that can successfully detect food fraud. Because new scientific papers list some techniques that are specific to some matrices, the list of techniques is described in a very limited way. It is therefore recommended to expand the number of laboratory analytical techniques, for food categories most described in this manuscript, also in tabular form.
- The Conclusions mention the scientific paper by Richter (32) from which the conclusions were misinterpreted - namely, the sentence "...reported the addition of honey and walnut honey to bee honey as one of the most common frauds with honey" is false (honewdew is more expensive than floral honey...) and not accurate since the summary of this paper (from 2011.) notes the opposite. Therefore, this sentence and this author are irrelevant and old source which need to be removed and replaced with a newer source and knowledge about honey as one of the most forged food categories.
- Lines 303 to 305 repeat the same text in lines 306 to 310 (Author's contributions)

Author Response

In accordance with the suggestion, the manuscript has been strongly revised.

Reviewer 3 Report

This paper is potentially interesting but there are some issues that should be carefully addressed by authors.

There are some major issues that should be addressed by authors.

Firstly, it is not understandable why the chosen period of analysis is 23 years, and why is 2004 the year of interest (pre and post).  To my knowledge, food fraud in Italy dominates among the other European Union Member States regarding the OPSON police actions.

''Food fraud cases in Italy: A 23-year analysis'' the topic does not suited the content of the article, because there is no instrumental analysis and the data of analysis.

There is no description of single case study of food fraud regarding the data that were presented in the article, which could increase the quality of the article. 

Line 72: It was not understandable ''hygiene package'' term which should be described more.

Line 98: NPC acronym is not explained.

Figure 1. does not have the citation, which are well known data that authors found in literature.

The data were collected from the Central Inspectorate of Quality Protection and Fraud Repression of Agri-food Products inspection activities but it is a pity that you do not have data on analytical methodology used in proving of food fraud or just several case studies regarding the inspection data.

The data are presented in 9 figures and later the same in tables. So the same data are shown several times.

Nine figures could be presented in only one figure, to get some more space. 

The data shown in the Tables is not understandable, every Table should be self explanatory. There are no measure units in the Tables.

Table 2, 3 and 4 are not understandable regarding measuring units and percentage. 

In Results, total amount of cases and number of cases per group could be helpful.

In Results, types of food fraud regarding the every group of the food should be considered.

Figures 10 and 11 do not have measuring units, and please explain word ranking.

Line 187 reduction in value: how did you calculate? Did you take into consideration the first and the last year?

Discussion section does not explain the data in Tables and Figures at all except the meat.

Line 244: criminal activities should be checked and explained with Italian Criminal law, or in recent European literature. 

Conclusion section has a few citations of other articles so I propose to delete citation from the Conclusion. Also, analytical techniques are not described previously.

Author Response

This paper is potentially interesting but there are some issues that should be carefully addressed by authors.

There are some major issues that should be addressed by authors.

  • Firstly, it is not understandable why the chosen period of analysis is 23 years, and why is 2004 the year of interest (pre and post). 

We have considered two periods (pre and post 2004) in consideration of the entry into force in 2004 of EC regulations so-called “hygiene package” and verify its effectiveness.

  • To my knowledge, food fraud in Italy dominates among the other European Union Member States regarding the OPSON police actions.

Obviously, Italy has participated at EU level with other countries in OPSON police actions. But data observed in our study concern irregularities discovered solely by ICQRF.

  • ''Food fraud cases in Italy: A 23-year analysis'' the topic does not suited the content of the article, because there is no instrumental analysis and the data of analysis.

We are not agree because data of analysis are present. Anyway we have changed the title as follows:

The “hygiene package”: analysis on fraud rates in Italy in the period before and after its entry into force

  • There is no description of single case study of food fraud regarding the data that were presented in the article, which could increase the quality of the article. 

We have added some examples reported in reports of ICQRF

  • Line 72: It was not understandable ''hygiene package'' term which should be described more.

The so-called Hygiene Package is the EU’s legal linchpin for ensuring the production of safe food within and outside the Union. When referring to the Hygiene Package account is taken of Regulations EC 852/2004, 853/2004, 854/2004 and 882/2004.

  • Line 98: NPC acronym is not explained.

We have specified the acronym in the text (NPC = Non Parametric Combination)

  • Figure 1. does not have the citation, which are well known data that authors found in literature.

The idea of this figure was born by visiting the following website https://knowledge4policy.ec.europa.eu/food-fraud-quality/topic/food-fraud_en

  • The data were collected from the Central Inspectorate of Quality Protection and Fraud Repression of Agri-food Products inspection activities but it is a pity that you do not have data on analytical methodology used in proving of food fraud or just several case studies regarding the inspection data.

In consideration of your observation, we have added a sentence in order to clarify this limit.

  • The data are presented in 9 figures and later the same in tables. So the same data are shown several times.

Nine figures could be presented in only one figure, to get some more space.

Figures 1-9 have been presented in a single figure (Figure 2)

  • The data shown in the Tables is not understandable, every Table should be self explanatory. There are no measure units in the Tables.

Table 2, 3 and 4 are not understandable regarding measuring units and percentage.

Table 2 (now table 4) reports mean, standard deviation and median of the fraud rates that were obtained from the ratio between the number of irregularities and the total inspections (part-to-whole ratios). The p-value obtained from the application of the NPC procedure is also reported

Table 3 (now table 5) reports the order place obtained by NPC ranking application and related to the fraud rates in the different sectors recorded before and after 2004

Table 4 (now table 6) reports, for every sector, the trend index; it is a measure that summarizes all the dispute rates recorded in all examined years.

None of the tables show percentage values.

In Results, total amount of cases and number of cases per group could be helpful.

  • In Results, types of food fraud regarding the every group of the food should be considered.

We have added some of the irregularities reported in ICQRF reports

  • Figures 10 and 11 do not have measuring units, and please explain word ranking.

Figure 10 (now figure 3) and Figure 11 (now figure 4) show the values obtained from the application of the Fisher combination function (fundamental tool in the NPC Ranking), which summarizes the fraud rates in the years before and after 2004, separately. The result values of Fisher combination function are expressed in the same unit of measure as the fraud rates (part-to-whole ratios).

  • Line 187 reduction in value: how did you calculate? Did you take into consideration the first and the last year?

The reduction in value indicates that, in the entire examined period (1997-2020), the fraud rate in this sector showed a decrease. It is calculated considering all the fraud rates detected from 1997 to 2020.

  • Discussion section does not explain the data in Tables and Figures at all except the meat.

For all the other sectors, given the reduction in the number of complaints observed, it is presumable that the legislation was effective; for this reason we did not consider it appropriate to take them into consideration.

About this, we have added a sentence at the end of the discussion.

  • Line 244: criminal activities should be checked and explained with Italian Criminal law, or in recent European literature. 

In this context it was not considered appropriate to refer to what criminal activities in Italy are attributable, but as emerges from the literature (Spink and Moyer, 2011) they may be associated with organized criminal structures such as the mafia, Cosa Nostra, Ndrangheta, etc.

Conclusion section has a few citations of other articles so I propose to delete citation from the Conclusion. Also, analytical techniques are not described previously.

Conclusion section has been modified. 

Round 2

Reviewer 1 Report

Authors made adequate changes in the manuscript.

Author Response

Dear reviewer, 

You will find below the final poof of the paper.

Thank you for your suggestion.

Reviewer 3 Report

Dear authors,

I accepted your response to my review sugestions. You have done great improvements regarding review.The manuscript deals with an interesting topic, but to me who does not understand the statistic terms very well, it is still difficult to follow the topic. So the Tables are not explained and understandable yet but if the editor is satisfied I don't have the problem.

Corrections:

1. Table 2. is completely unnecessary and is unrelated to the rest of the text. Perhaps, if these were the cases in Italy it would be related to the subject of the manuscript. This Table should be related the topic or just erase the Table 2. 

2. Total number of cases is not written.

3. Sorry to hear that you could not be able to refer to Italian Criminal Law, because of the food criminal activities that could be in relation with mafia. 

It could be done with enumeration and description of felonies that could be commited with food regarding the Italian Criminal Law without pointing to anyone, especially  mafia, or corrupted police officers. 

The main problem is the same in all European countries. No criminal prosecutions yet. 

Author Response

Dear Reviewer,

Thank you very much for your time and all your comments.

We have revised the manuscript considering your comments; the answers to your questions are given below.

The changes made in the manuscript to address comments are marked up using the “Track Changes” function.

Corrections:

  1. Table 2. is completely unnecessary and is unrelated to the rest of the text. Perhaps, if these were the cases in Italy it would be related to the subject of the manuscript. This Table should be related the topic or just erase the Table 2. 

R. We have deleted the table 2

  1. Total number of cases is not written.

R. The total number of inspections and irregularities has been included. Also a table with detailed information has been added.

  1. Sorry to hear that you could not be able to refer to Italian Criminal Law, because of the food criminal activities that could be in relation with mafia. 

R. It could be done with enumeration and description of felonies that could be committed with food regarding the Italian Criminal Law without pointing to anyone, especially mafia, or corrupted police officers. 

The main problem is the same in all European countries. No criminal prosecutions yet. 

We have inserted more sentences about that.
